# Symptoms of Anxiety and Depression in Italian Nursing Students: Prevalence and Predictors

**DOI:** 10.3390/healthcare12212154

**Published:** 2024-10-29

**Authors:** Felice Curcio, Lara Guedes de Pinho, Cristiana Rago, Davide Bartoli, Gianluca Pucciarelli, Cesar Iván Avilés-González

**Affiliations:** 1Faculty of Medicine and Surgery, University of Sassari (UNISS), 07100 Sassari, Italy; 2Department of Nursing, University of Évora, 7002-554 Évora, Portugal; lmgp@uevora.pt; 3Department of Biomedicine and Prevention, University Tor Vergata, 00133 Rome, Italy; ragocristiana@gmail.com (C.R.); bartoli.davide90@gmail.com (D.B.); gianluca.pucciarelli@uniroma2.it (G.P.); 4Department of Medical Sciences and Public Health, University of Cagliari, 09042 Cagliari, Italy; cesari.avilesg@unica.it

**Keywords:** anxiety, depression, nursing students, psychological distress

## Abstract

**Background/Objectives**: Mental disorders in nursing students, although very common under normal circumstances, have worsened over time due to the COVID-19 pandemic. This study aimed to assess (1) what the prevalence of anxiety and depression symptoms in Italian nursing students was and (2) what factors were associated with them. **Methods**: In May 2023, a cross-sectional study was conducted on the bachelor’s degree Nursing course in an Italian university. Levels of anxiety and stress were assessed using the Generalized Anxiety Disorder Scale (GAD-7) and the Patient Health Questionnaire (PHQ-9), respectively. In addition, socio-demographic variables and data on mental health, drug intake, and substance use were collected. Multiple binary logistic regression adjusted analyses were used to identify predictive factors. **Results**: A total of 148 nursing students completed the questionnaire. A total of 9.5% reported a previous diagnosis of a mental disorder, and, of these, 35.7% reported that it was diagnosed after the start of the COVID-19 pandemic. The mean GAD-7 and PHQ-9 score was 9.68 (SD = 5.2) and 8.37 (SD = 5.6), respectively. Protective factors for depressive symptoms included not having a previous diagnosis of a mental disorder (adjusted odds ratio = 0.10, 95% CI: 0.002–0.47, *p* < 0.001) and a perceived high (adjusted odds ratio = 0.03, 95% CI: 0.003–0.22 *p* < 0.001) or medium (adjusted odds ratio = 0.14, 95% CI: 0.03–0.82, *p* < 0.05) socioeconomic level, while the predictor of anxiety symptoms was returning home once a month for students studying away from their residence (adjusted odds ratio = 6.4, 95% CI: 1.01–40.8, *p* < 0.05). **Conclusions**: Urgent investments are needed in universities to implement mental health promotion programmes and to offer counselling services to reduce and prevent mental health problems among students.

## 1. Introduction

Beginning university studies is a complex transition process that subjects students to numerous sources of stress, both academic and social [1,2]. At this stage of life, students are subjected to academic stressors, such as an increased workload and lower-than-expected grades; interpersonal stressors, such as exposure to new people, having to work with people they do not know, and the desire to make a good impression on others; and intrapersonal stressors, such as new responsibilities, being in a non-domestic environment for the first time, changes in eating and sleeping habits, and managing finances and time between university, social relations, and leisure time [2,3,4,5,6]. These difficulties tend to lead to permanent states or high levels of stress that compromise the well-being and academic engagement of university students [7,8]. Moreover, several studies have reported that most mental disorders manifest themselves in the young adult stage, with their onset before the age of 24, so university students are particularly vulnerable [9,10,11].

In the literature, it is well documented that the area of training influences levels of mental distress, such as anxiety and depression [12]; higher rates have been reported in health area students than in students from other areas [13,14]. In addition to theoretical learning, which is common to all students, nursing students have to undertake clinical practice placements in the nursing environment, which is an additional stress factor for students [15,16]. Higher levels of anxiety during the clinical training of nursing students have frequently been reported, which are mainly due to new or unfamiliar clinical settings and the fear of committing mistakes, causing harm to patients, and experiencing negative reactions [17,18]. In turn, university programmes require nursing students to acquire high levels of knowledge and skills in clinical practice [19,20]. Furthermore, nursing students must continually deal with the suffering, pain, or death of patients [21]. Over the years, nursing has achieved a holistic approach, which goes beyond the care of the individual and addresses the environment, the person, the family, the community, and health [22,23,24]. This has led to nurses being the health professionals who consistently spend the most time caring for and educating patients.

In 2020, with the advent of the COVID-19 pandemic, changes in population health occurred worldwide. Several studies report that the younger population, including students, suffered the most from mental disorders such as anxiety and depression [11,25]. The main measures taken have focused on the prevention and containment of illness by limiting physical and social contacts. In Italy, for example, all universities stopped academic activities such as lessons, laboratories, and internships, as well as closed university facilities. This decision was taken in order to eliminate direct contact among people and minimise transmission between people from different areas [26]. The context experienced during COVID-19, such as the lockdown, required a readjustment of both the education system and university students, negatively impacting the development and/or increase in mental disorders globally [27,28,29,30].

In light of the above, it is crucial to understand and identify high levels of post COVID-19 anxiety and depression in Italian nursing students at an early stage in order to protect their mental health. This will also help to provide a basis for the development of mental health prevention and promotion programmes. Therefore, the aim of this study is to assess the prevalence of anxiety and depression among Italian nursing students and to analyse the factors associated with these symptoms. To this end, the following research questions were formulated:(1)Among Italian nursing students, what is the prevalence of anxiety and depressive symptoms?(2)What are the factors associated with anxious and depressive symptoms in Italian nursing students?

## 2. Materials and Methods

### 2.1. Study Design

A cross-sectional study was conducted among students of the bachelor’s degree Nursing course in the University of Sassari, Sardinia, Italy. This study was conducted according to the STROBE (STrengthening the Reporting of OBservational studies in Epidemiology) statement [31].

### 2.2. Participants and Procedure

Nursing students at the University of Sassari (Italy) completed an online questionnaire investigating mental disorders and their associated factors. Non-probability convenience sampling was used. The total number of students enrolled in the bachelor’s degree course in Nursing was 422 students.

The inclusion criteria for the sample were (1) being a student enrolled in a nursing degree programme and (2) having access to the Internet. The exclusion criteria were (1) being an international student on study mobility.

The online questionnaire was constructed using the free-access Google Forms application, and the questionnaire link was sent through the students’ WhatsApp groups. The students were asked to complete the questionnaire in their classrooms. The average time to complete the questionnaire was around 10 min. Data collection took place in May 2023.

This study was conducted in accordance with the Declaration of Helsinki, the Italian privacy law (Decree No. 196/2003), and the General Data Protection Regulation (GDPR-EU 2016/679) and was approved by the Faculty Director of the participating university (date of approval 20 December 2022). Informed consent was obtained from students before to complete the questionnaire. Students were informed that their participation was voluntary and anonymous. In addition, students were informed that they could leave the study at any time without any adverse consequences for their university programme.

### 2.3. Instruments

The questionnaire completed by the nursing students contained 29 elements and was divided into four sections. In the first section, there were items that collected information such as age, sex, emotional relationship, nationality, perceived socioeconomic level, and frequency of going to the official residence. The second section included questions regarding (1) the year of the course attended, (2) students’ perceptions of their academic performance, and (3) student worker status, with quantification of the number of hours worked per week. The third session collected the following data: (1) consumption of coffee, consumption of alcoholic beverages, cigarette smoking, and drug use; and (2) the frequency of consumption and (3) the quantity consumed. Participants also reported on their perceptions of how COVID-19 affected their mental well-being; whether they had been diagnosed with mental disorders and if they predated the advent of COVID-19; if they had seen a mental health specialist; and if they consumed drugs due to mental disorders, what kind of medications they took, and whether they were prescribed by a doctor. The last section of the questionnaire assessed symptoms of mental disorders, such as (a) symptoms of anxiety with the Generalized Anxiety Disorder Scale (GAD-7) [32] and (b) symptoms of depression with the Patient Health Questionnaire (PHQ-9) [33].

The GAD-7, developed by Spitzer et al., was used to evaluate how often anxiety-like symptoms occurred in the last 2 weeks, with symptoms such as (1) feeling nervous, anxious, or agitated; (2) not being able to stop or control worrying; (3) worrying too much for various reasons; (4) trouble relaxing; (5) being so tense that it is hard to sit still; (6) being easily annoyed or irritable; and (7) feeling frightened as if something bad might occur. Respondents’ answers were collected on a 4-point Likert scale (0  =  not at all, 1  =  some days, 2  =  more over half the days, and 3  =  almost every day), with total scores ranging from 0 to 21 (0–4 = anxiety minimal, 5–9 = mild anxiety, 10–14 = moderate anxiety, and 15–21 = severe anxiety). The GAD-7 showed adequate psychometric characteristics (specificity, sensitivity) and internal consistency, with values exceeding 0.80 [34,35]. In our sample, the internal consistency was α = 0.89.

The PHQ-9, developed by Kroenke et al. in 2001 [33], was used to identify how often depressive symptoms occurred in the last 2 weeks. Respondents’ answers were collected on a 4-point Likert scale (0  =  not at all, 1  =  some days, 2  =  more over half the days, and 3  =  almost all day), with total scores ranging from 0 to 27. Higher PHQ-9 scores suggest the presence of more severe depressive symptoms (5–9 = mild depression, 10–14 = moderate depression, 15–19 = moderately severe depression, and 20–27 = serious depression). The internal reliability of the PHQ-9 is excellent, with a Cronbach’s alpha greater than 0.80 [36]. In our sample, the internal consistency was α = 0.88.

### 2.4. Statistical Analysis

A descriptive analysis of the variables was performed, expressing the qualitative variables in frequencies and percentages and the quantitative variables in means or medians and standard deviations or interquartile ranges.

Spearman’s correlation coefficient was used to measure the correlation between age and the GAD-7 and PHQ-9 scale scores, whereas Kendall’s correlation coefficient was used to measure the correlation between frequency of substance use and the items of the GAD-7 and PHQ-9 scales.

The Shapiro–Wilk test and the analysis of skewness and kurtosis coefficients were used to assess the normality hypothesis of the numerical variables. Levene’s test was used to assess the homogeneity of variances. 

Due to the violation of the normality hypothesis, the Mann–Whitney–Wilcoxon test was used to compare the total scale scores between two groups (e.g., gender, transferred from official residence, etc.), whereas the Kruskal–Wallis test was used to compare total scale scores between more than two groups (e.g., perception of educational achievement, perception of socioeconomic level, etc.). The effect size was calculated using the biserial rank correlation (1), recommended for estimating effect size in nonparametric tests [37], and the better-known Cohen’s d (2) (0.2 = small effect, 0.5 = medium effect and 0.8 = great effect) [38], which is used to calculate effect size in parametric tests.

Multiple binary logistic regression was used to examine the predictor variables of anxiety and depression levels, reporting the results as adjusted odds ratios (adjusted ORs) and their respective 95% confidence intervals (CIs). The GAD-7 and PHQ-9 scale scores were dichotomised, respectively, as high levels of symptoms (scores ≥ 10) and low levels of symptoms (<10). Finally, the goodness-of-fit indices (Hosmer–Lemeshow test, r2 Cox–Snell, r2 Nagelkerke) and the predictive accuracy (sensitivity, specificity, validity index) of the predictive models of anxiety and depression obtained were verified.

Analyses were performed using IBM© SPSS Statistics v.25.0 and the statistical package Jamovi v.2.4. The significance level was set at 0.05.

## 3. Results

### 3.1. Socio-Demographic and Academic Characteristics

The survey was sent to 422 nursing students. Among them, 148 (35.7%) completed the questionnaire (35.7%). The average age of the nursing students was 24.81 (±6.30), with the youngest being 19 and the oldest being 51. The majority of the participants were female (82.4%). A total of 43.2% of the students lived far from home, with an average distance between their place of study and their residence of about 127.6 (SD = 62.7) km. Most of the participants were not working students (83.3%); 55.40% of them received financial support for their studies/living expenses from their parents, 45.3% received a study grant, and 4.3% received state support. The other socio-demographic and academic characteristics are explained in Table 1.

### 3.2. Mental Health Outcomes

The mean GAD-7 score was 9.68 (SD = 5.2), and 83.8% of participants reported mild to severe anxiety symptoms. Overall, 27% (n = 40) of the students reported minimal levels of anxiety disorders, 37.2% (n = 55) reported mild levels, and 16.2% (n = 24) reported minimal levels, whereas severe levels of anxiety were reported by about 1/5 of the students (19.6%; n = 29) (Figure 1). In the 2 weeks prior to the completion of the survey, the problems that participants reported most often (at least more than half of the days) were “Worrying too much about different problems” (56.8%), “Not being able to stop worrying or keeping worries under control” (47.3%), “Having difficulty relaxing” (47.3%), “Feeling nervous, anxious or irritable” (43.9%), and “Getting annoyed or irritated easily” (41.2%). “Being afraid that something terrible will happen” and “Being so restless that you find it hard to sit still” were the least frequently reported problems, 27% and 16.9%, respectively.

With regard to depressive symptoms, the mean score of the PHQ-9 scores was 8.37 (SD = 5.6). Most students (32.4%, n = 48) reported mild levels of depression, compared to 25.7% (n = 38), who reported moderate levels, and 9.5% (n = 14), who reported moderately severe levels. However, 4.1% (n = 6) of the students showed scores indicative of severe depression (Figure 2). In the 2 weeks prior to the completion of the survey, the problems that participants reported most often were “Feeling tired or low energy” (56.1%), “Suffering from insomnia or sleeping too much” (37.2), “Feeling down, bad mood” (35.8%), and “Difficulty in concentrating” (27.7%). The remaining items of the Patient Health Questionnaire (“Lack of interest or pleasure in doing things”; “Having poor appetite or losing weight”; “Feeling annoyed, let down or feeling abandoned by the family”; “Moving or talking so slowly that others do not understand you or feeling agitated and moving more than usual”; “Thinking that it would be better to die or hurt yourself in some way”) were reported by less than a quarter of the students.

When asked “Have you ever been diagnosed with any kind of mental disorder?”, the vast majority of students answered no; however, 9.5% (n = 14) reported being diagnosed with a mental disorder, and, of these, 35.7% (n = 5) reported being diagnosed after the start of the COVID-19 pandemic. The diagnoses most frequently self-reported by students, and which had previously been diagnosed by a specialist, were anxiety and depression, 6.1% and 5.4%, respectively, with 6.8% of students reporting two or more diagnoses. When asked “How has COVID-19 affected your mental health?”, 45.9% reported that it had no impact. However, 39.2% reported that their mental health from 2020 to 2022 deteriorated, and 10.1% reported that it had severely deteriorated.

With regard to taking drugs, 26.9% of the participants reported taking them for some mental problem. Of these, 19.6% took natural drugs (Valerian, other) while 3.4% took antidepressants, and another 3.4% took benzodiazepines. However, only 12% reported that the medication had not been prescribed by a doctor, and 9.5% of the students reported that they had received at least one psychiatric consultation. Finally, with regard to the problems mentioned above, 58.8% of the students stated that these disorders caused few difficulties in daily life, 12.8% of the sample reported many difficulties, and only 25.7% reported no difficulties.

### 3.3. Substance Use

The substance most commonly consumed by the students was coffee or other caffeinated beverages (93.9%), followed by alcohol (62.8%), tobacco (43.2%), and cannabis (11.5%). The average weekly intake of alcoholic beverages reported by participants was in 59.5% of cases between 1 and 7 drinks per week. However, 58.1% of the students reported consuming four to five servings of alcohol on the same occasion.

### 3.4. Factors Associated with Depression and Anxiety

No significant correlation was found between age and the Generalised Anxiety Disorder scale (GAD-7) (r = −0.037; *p* = 0.660) or the Patient Health Questionnaire (PHQ-9) (r = −0.033; *p* = 0.691).

Table 2 shows the differences observed through the Mann–Whitney–Wilcoxon or Kruskal–Wallis tests. The dependent variables used in the analyses were the GAD-7 and PHQ-9 scores, while the independent variables included socio-demographic characteristics, such as age, sex, etc.; students’ perceptions of their own academic performance and socioeconomic level; student worker status; and consumption habits, such as cigarette smoking, cannabis use, alcoholic beverage consumption, etc. As can be observed, significantly higher depressive symptom scores were encountered in students who were female (*p* = 0.037), had a low perceived socioeconomic level (*p* = 0.024), had a previous diagnosis of mental disorders (*p* = 0.006), had a low perceived academic performance (*p* = 0.001), smoked tobacco (*p* = 0.008), used cannabis (*p* = 0.034), and consumed more than 4–5 alcoholic drinks on the same occasion (*p* = 0.025). With regard to anxiety symptoms, they were significantly higher in students in a loving relationship (*p* = 0.043), with a previous diagnosis of mental disorders (*p* = 0.014), with a perceived low school performance (*p* = 0.048), who smoked tobacco (*p* = 0.032), and who consumed more than 4–5 alcoholic drinks on the same occasion (*p* = 0.009).

In relation to symptoms of depression, the multiple binary logistic regression (Table 3) model showed that “not having a previous diagnosis of a mental disorder” reduced the probability of developing depressive symptoms by 90% (adjusted odds ratio = 0.10, 95% CI: 0.002–0.47, *p* < 0.001) compared to those who had received a previous diagnosis. Similarly, a high (adjusted odds ratio = 0.03, 95% CI: 0.003–0.22 *p* < 0.001) or medium (adjusted odds ratio = 0.14, 95% CI: 0.03–0.82, *p* < 0.05) perceived socioeconomic level reduced the likelihood of suffering from depressive symptoms. This model showed a goodness-of-fit index of 0.21 (r2 Cox-Snell = 0.21) and 0.29 (r2 Nagelkerke 0.29). Finally, the model has the ability to correctly classify 70.9% of the cases, better classifying students without depressive symptoms.

Table 4 shows the results of the logistic regression model on the predictors of anxiety levels among Italian nursing students. For students studying away from their residence, returning home once a month compared to every week increases the probability of developing anxiety symptoms more than 6-fold (adjusted odds ratio = 6.4, 95% CI: 1.01–40.8, *p* <0.05). This model has the ability to correctly classify 67.2% of the cases, with a sensitivity and specificity of 72.7% and 61.3%, respectively. The goodness-of-fit indices were the Hosmer–Lemeshow test = 0.987; r2 Cox–Snell = 0.155; r2 Nagelkerke = 0.2.

## 4. Discussion

Our study found that a significant percentage of Italian nursing students had symptoms of anxiety and depression. Specifically, 46.6% of the participants reported symptoms of moderate to severe anxiety according to the GAD-7 scale. With regard to depressive symptoms, however, 71.7% of the students indicated the presence of mild to severe depressive symptoms.

Although there are no studies on pre-pandemic levels of anxiety and depression in Italian nursing students, in our study we found a higher prevalence of anxiety symptoms compared to studies in other contexts. For example, Reverté-Villarroya et al. [39], in their 2017 study conducted in Spain, reported that 35.9% of nursing students had a GAD-7 score above 10 (a threshold defined for moderate levels of anxiety). Similarly, Kim et al. [40] reported that 19.4% of the students surveyed scored greater than or equal to 10 on the GAD-7 scale, whereas a study conducted in Italy during the pandemic (17 March to 27 March 2021) found that 46.1% of nursing students reported moderate to severe levels of an anxiety disorder [29]. Another survey of 244 nursing students conducted during the third week of lockdown also reported that 42.8% had moderate to severe levels of anxiety [41]. Both of these studies reported results overlapping with ours, unlike Zhu’s study which reported anxiety levels in 10.1%, after the COVID-19 outbreak, between moderate and severe [15].

Regarding levels of depression, the results were similar; although the data should be analysed with caution, as different scales were used. Studies conducted in the pre-pandemic period with nursing students reported lower levels of depression than our results, ranging from 28.7% in a study of Chinese nursing students [42] to 38.7% in Iranian students (data collected in 2012) [43]. A study conducted in Italy, from 25 March to 25 April 2020, reported similar results to ours, although mild to severe depressive symptoms were slightly higher (71.7% vs. 81.40%) [44]. Similarly, a study of Spanish nursing students, one year after the COVID-19 pandemic, reported that 81% of participants suffered from depressive symptoms [45]. In contrast to our results, a study by David et al. [46], conducted on Romanian nursing students, reported that 37.78% of them had depressive symptoms.

Finally, examining a recent meta-analysis that included 20 studies, with a total of 7742 nursing students, it was found that the anxiety scores of nursing students during the COVID-19 pandemic were slightly higher (50%) than before the pandemic [11]. Similarly, a systematic review with meta-analysis by Quesada-Puga et al., in which 14 studies were included, concluded that there was a high prevalence of depression among young university students, with figures exceeding 50% [47]. These studies conducted in different geographical contexts, combined with those conducted in Italy, consolidate the evidence that, globally, the COVID-19 pandemic had a negative impact on nursing students’ mental health (depression and anxiety). Social restrictions, academic interruptions, and financial uncertainty during the COVID-19 pandemic are among the main factors to be implicated in the worsening mental health of students [48,49].

The results of this study are consistent with the previous literature that has shown that socio-demographic factors such as gender, socioeconomic status, and academic performance have a significant association with levels of anxiety and depression in nursing students. Specifically, women reported significantly higher levels of depression than men (*p* = 0.037), which is in line with research indicating that the female gender tends to be more vulnerable to anxiety and depression-related mental health conditions [50,51,52]. For example, a study by Huang et al. [53], conducted exclusively on female nursing students in Taiwan, showed that body image perception and body mass index (BMI) significantly influence the mental health of these students. This study found that 45.2% of the participants had mental disorders, highlighting the relationship between perceived stress and mental health problems related to self-image, which is particularly relevant in women. Similarly, Yamazaki et al. [54] evaluated the effectiveness of habitual exercise on health promotion in university students, observing significant improvements in body composition and mental health after eight weeks of regular exercise. These results suggest that exercise is an effective tool for improving mental health and reducing stress levels in women.

Socioeconomic status also appears to play an important role in the severity of depressive symptoms. Students with low perceived socioeconomic status show significantly higher scores on the PHQ-9 than students with medium and high perceived socioeconomic status (*p* = 0.024). This association may be related to the greater economic difficulties faced by low-income students; they have to balance academic responsibilities with financial concerns, exacerbating their vulnerability to mental disorders such as depression.

In line with our findings, previous studies have shown that mental health significantly influences the academic performance of nursing students. According to Atashzadeh-Shoorideh et al. [55], the implementation of the “clinical competence model for mental health” significantly improved students’ academic performance by developing specialised clinical skills. On the other hand, Oliveira Silva et al. [56] found that 60.2% of students had depressive symptoms, which negatively affected their school performance and increased their risk of school failure and dropout. Finally, Fauzi et al. [57] highlight that stress, anxiety, and depression in health science students are directly related to a decrease in their academic performance, emphasising the need for psychosocial interventions to improve well-being and academic success in nursing programmes.

Students with consumption habits, such as smoking tobacco and using cannabis, showed higher levels of depressive symptoms than those without these consumption habits. Furthermore, tobacco use among nursing students is associated with higher levels of anxiety (*p* = 0.032). Our results are in line with studies in the literature. A study with a sample of Spanish nursing students showed that students who smoked tobacco had a 30% increased risk of suffering from depressive symptoms and a 63% increased risk of anxiety disorders. In addition, regular alcohol consumption (≥2 times per week) was strongly associated with anxiety states [58]. A study conducted between March and December 2018 among university students on 10 Ukrainian campuses showed that participants were more likely to report depressive symptoms if they were involved in alcohol and cannabis use, while, contrary to our findings, tobacco use was not significantly associated with depressive symptoms [59]. A study by Ramón-Arbués et al. [60] also showed smoking tobacco and habitual alcohol consumption were significantly associated with anxiety symptoms among university students. Finally, the results of the study by Blows and Isaacs [61] show that most of the students interviewed started using substances (both alcohol and cannabis) only after entering university. Among the reasons given by students to justify these drinking habits, we find mainly difficulties in emotional, social, and family relationships, financial pressures, academic performance improvement, and personal stress management [62]. However, these habits often have negative effects on mental health, contributing to a vicious cycle that results in anxiety symptoms and depression [63].

The results of our study show that predictive factors for anxiety and depressive symptoms are a previous diagnosis of a mental disorder, a low socioeconomic level, and going home once a month. Comparable results were also found in other studies. Amaro et al. [64], in their study of Spanish university students, reported that those who lived away from their families and returned home less frequently developed higher levels of anxiety and depression than those who returned home more frequently (once a week). Another study, conducted in Ethiopia on university medical students, also reported that students who studied away from their residence suffered more from mental disorders [65]. This could be explained by the importance of the family unit as the main source of emotional support in students, as they face numerous sources of stress [66].

In line with our findings, previous studies suggest that students with low financial support often experience higher levels of stress and worry, which can trigger or exacerbate mental health problems. Li et al. [67], in their systematic review and meta-analysis, also confirm a higher prevalence of depressive symptoms and anxiety in university students with low socioeconomic status. Another study, conducted during the closure of universities due to the COVID-19 pandemic, showed that students with low socioeconomic status reported a higher level of psychological distress and loneliness [68]. By contrast, students with higher socioeconomic status tended to have higher life satisfaction and emotional well-being. As suggested by a UK study [69], these effects are mediated by perceived social support and self-efficacy. In addition, financial problems affecting students may influence some of them to drop out of school, which worsens their mental state [70]. Also Agyapong-Opoku et al. [71] showed that students with lower socioeconomic status and financial problems exhibited higher levels of depression and anxiety due to the stress generated by financial and academic burdens. These findings reinforce the relationship between lower socioeconomic status and worse mental health in university students.

This study presents some limitations that should be discussed. First, the absence of a control group did not make it possible to determine whether nursing students have a higher prevalence of anxiety and depression symptoms compared to the general population. Furthermore, we used a convenience sample from only one Italian university. This reduces the generalisability of the results to all Italian nursing students, thus reducing the study’s external validity. Second, a low adherence to filling in the questionnaires by the students was observed. This could potentially have influenced the results of our study; therefore, further studies with a larger sample are needed. Third, the data were collected using a self-administered questionnaire; therefore, an assessment bias may be present. However, the variables analysed in this study refer to students’ perceptions of their mental health and should not be interpreted as a means of clinical diagnosis of mental disorders. In this way, collecting data using self-reported measures is adequate for perception data. Finally, as this was a cross-sectional research study, it was not possible to establish causal relationships between the COVID-19 pandemic and the observed variables, let alone determine causal effects between the variables. Longitudinal studies will be needed in the future to assess how anxiety and depression levels fluctuate over time in response to different stressors. Furthermore, the coping strategies used and the impact of actions to promote the mental health of nursing students will need to be evaluated.

Despite the limitations mentioned above, on the whole, our results provide an overview of the mental health status of Italian nursing students, in itself an already vulnerable population, which has experienced constant changes related to the COVID-19 pandemic and a tendency to increase mental disorders. We hope that our results may help to implement psychological support programmes specifically aimed at nursing students, who seem to be more at risk due to the academic demands and stress resulting from their training. Universities should develop strategies that promote stress management and the accessibility of mental health services for students. Furthermore, it is essential to address socioeconomic inequalities within the academic environment by offering financial and emotional support to students in need.

## 5. Conclusions

This study explored mental disorders in Italian nursing students at a time when almost all restrictions related to the COVID-19 pandemic had lapsed. Our results revealed a high prevalence of anxiety and depression symptoms among these students. Factors such as romantic relationships, previous diagnosis of mental disorders, perceived academic performance, tobacco smoking, cannabis use, and drinking more than 4–5 alcoholic beverages on the same occasion showed an association with depressive and/or anxiety symptoms. Furthermore, women and students with a low perceived socioeconomic level were more likely to experience high levels of depressive symptoms. However, the most important predictors of depressive symptoms were a previous diagnosis of a mental disorder and a low socioeconomic level, while going home once a month increases the likelihood of developing anxiety symptoms by 6 times.

Overall, the results indicate that the COVID-19 pandemic is likely to have a clear impact on the mental health of nursing students. Furthermore, the high consumption of substances such as coffee, alcohol, and tobacco among students suggests that many of them use these remedies as coping strategies to deal with academic and personal challenges. However, these habits can exacerbate mental health problems, as shown by the association between tobacco use and high levels of anxiety.

There is an urgent need for policy makers, university stakeholders, health services, and society at large to invest in implementing mental health promotion programmes and offering counselling services to reduce and prevent mental health problems among students.

## Figures and Tables

**Figure 1 healthcare-12-02154-f001:**
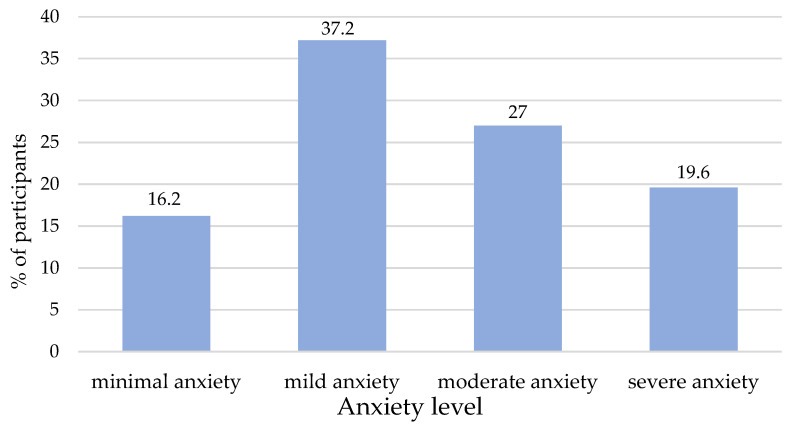
Levels of anxiety symptoms reported by Italian nursing students (n = 148).

**Figure 2 healthcare-12-02154-f002:**
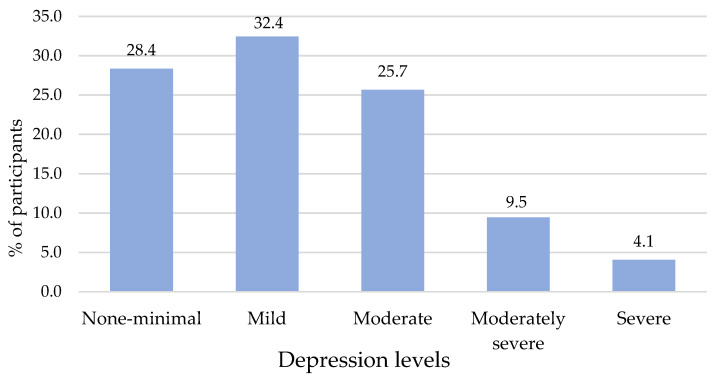
Levels of depressive symptoms reported by Italian nursing students (n = 148).

**Table 1 healthcare-12-02154-t001:** Socio-demographic and academic characteristics (n = 148).

Variables	Frequency (%)	Mean (SD)
Sex		
Male	26 (17.6)	
Female	122 (82.4)	
Age		24.8 ± 6.30
Nationality		
Italian	148 (100)	
Other	0 (0)	
Loving relationship		
Single	88 (59.5)	
Married(a)/Convivant	12 (8.1)	
With a partner but not living together	48 (32.4)	
Socioeconomic level (students’ perceptions)		
Low	7 (4.7)	
Medium	115 (77.7)	
High	26 (17.6)	
Relocated from the official residence		
Yes	64 (43.2)	
No	84 (56.8)	
If displaced, how often do you go home?		
Every weekend	19 (29.7)	
Two to three times a month	24 (37.5)	
Once a month	11 (17.2)	
Only during school breaks/school holidays	10 (15.6)	
Who you live with during your studies		
Parents	48 (32.4)	
Other relatives	8 (5.4)	
Study colleagues/friends	57 (38.5)	
Alone	12 (8.1)	
Boyfriend or girlfriend	13 (8.8)	
Other	10 (6.8)	
Working student		
Yes	24 (16.2)	
No	124 (83.8)	
How many hours do you work on average per week?		24.6 ± 15.7
Academic performance (student’s perceptions)		
Low	10 (6.8)	
Discrete	24 (16.2)	
Average	54 (36.5)	
Good	57(38.5)	
Excellent	3 (2.0)	

**Table 2 healthcare-12-02154-t002:** Association between socio-demographic variables, academic performance, substance use, and levels of anxiety and depression in Italian nursing students.

Variables	PHQ-9			GAD-7		
	ME (IQR)	*p*	Effect Size (1, 2)	ME (IQR)	*p*	Effect Size (1, 2)
Sex		**0.037** a	**0.26 (1)** **−0.42 (2)**		0.189 a	0.16 (1) −0.30 (2)
Male	4.50 (7.5)			8 (7.25)		
Female	8 (7)			9 (9)		
Loving relationship		0.560 a	0.06 (1) −0.1(2)		**0.043** a	**0.20 (1)** **−0.36 (2)**
In a relationship	7 (8.75)			11 (9)		
Not in a relationship	7.50 (7)			8 (7)		
Working student		0.802 a	0.03 (1) −0.16 (2)		0.427 a	0.10 (1) −0.18 (2)
Yes	8 (5.75)			8 (7.75)		
No	7 (8)			9 (8)		
Socioeconomic level (students’ perceptions)		**0.024** b	−		0.052 b	−
Low	14 (5)			13 (7)		
Medium	7 (8)			9 (8)		
High	5 (5.25)			8.50 (10)		
Relocated from the official residence		0.245 a	0.11 (1) 0.22 (2)		0.421 a	0.01 (1) 0.15 (2)
Yes	8 (8.75)			10 (8)		
No	7 (9)			8.50 (8)		
If displaced, how often do you go home?		0.088 b	−		0.112 b	−
Every weekend	5 (8)			8 (9)		
Two to three times a month	8 (7.5)			9 (8.5)		
Once a month	13 (9)			14 (4)		
Only during school breaks/school holidays	6 (11.5)			6 (6)		
Previous diagnosis of mental disorders		**0.006** a	**0.44 (1)** **1.14 (2)**		**0.014** a	**0.4 (1)** **0.74 (2)**
Yes	11 (12.5)			13.5 (8.5)		
No	6.5 (7)			8 (8)		
Academic performance (student’s perceptions)		**0.001** b	−		**0.048** b	−
Low	15.5 (10)			15.5 (9.75)		
Discrete	9 (6.75)			10.5 (8.25)		
Average	9 (9)			10.5 (9)		
Good	5 (4)			8 (7)		
Excellent	5 (5)			9 (9)		
Drink coffee		0.066 a	0.37 (1) −0.59 (2)		0.121 a	0.31 (1) −0.58 (2)
Yes	8 (8)			9 (8)		
No	4 (8.5)			5 (12)		
Smoke tobacco		**0.008** a	**0.25 (1)** **−0.52 (2)**		**0.032** a	**0.21 (1)** **−0.38 (2)**
Yes	10 (9)			10 (8)		
No	6 (6)			8 (7)		
Use cannabis		**0.034** a	**0.31 (1)** **−0.67 (2)**		0.235 a	0.18 (1) −0.38 (2)
Yes	11 (11.5)			11 (9)		
No	7 (7)			9 (8)		
Drink alcoholic beverages		0.127 a	0.15 (1) −0.25 (2)		0.318 a	0.1 (1) −0.19 (2)
Yes	8 (8)			11 (9)		
No	6 (6)			10 (8)		
Drink more than 4–5 alcoholic drinks on the same occasion		**0.025** a			**0.009** a	
Yes	9 (7.25)		**0.21 (1)**	11 (8.25)		**0.25 (1)**
No	5.5 (7.25)		**−0.39 (2)**	7 (7)		**−0.45 (2)**

GAD-7: Generalized Anxiety Disorder Scale; IQR: interquartile range; ME: median; PHQ-9: Patient Health Questionnaire. a, Mann–Whitney–Wilcoxon test; b, Kruskal–Wallis test. (1) Effect size calculated by the rank-biserial correlations (non-parametric approach). (2) Effect size calculated by the Cohen d test (parametric approach). Bold: significant results (*p* < 0.05)

**Table 3 healthcare-12-02154-t003:** Multivariable binary logistic regression for depression levels. DEPRESSION high (PHQ-9 ≥ 10); DEPRESSION low (PHQ-9 < 10).

Variables	OR Adjusted CI 95%	*p*
Previous diagnosis of mental disorders		
Yes	1 (Reference)	
No	0.1 (0.002–0.47)	<0.01
Socioeconomic level (students’ perceptions)		
Low	1 (Reference)	
Medium	0.14 (0.03–0.82)	<0.05
High	0.03 (0.003–0.22)	<0.01
Use cannabis		
No	1 (Reference)	
Yes	2.6 (0.81–8.5)	0.1
Drink alcoholic beverages		
No	1 (Reference)	
Yes	1.9 (0.84–4.4)	0.12
Smoke tobacco		
No	1 (Reference)	
Yes	2 (0.94–4.5)	0.07

**Table 4 healthcare-12-02154-t004:** Multivariable binary logistic regression for anxiety levels. ANXIETY high (GAD-7 ≥ 10); ANXIETY low (GAD-7 < 10).

Variables	OR Adjusted CI 95%	*p*
If displaced, how often do you go home?		
Every weekend	1 (Reference)	
Two to three times a month	0.935 (0.27–3.3)	0.92
Once a month	6.4 (1.01–40.8)	<0.05
Only during school breaks/school holidays	0.71 (0.14–3.7)	0.96
Loving relationship		
No relationship	1 (Reference)	
In a relationship	2.96 (0.95–8.9)	0.05
Previous diagnosis of mental disorders		
Yes	1 (Reference)	
No	0.275 (0.05–1.7)	0.16

## Data Availability

The raw data supporting the conclusions of this article will be made available by the authors on request.

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
