# Peer review of "Symptoms of Anxiety and Depression in Italian Nursing Students: Prevalence and Predictors"

_healthcare, 2024, doi:10.3390/healthcare12212154_

Round 1
Reviewer 1 Report
Comments and Suggestions for Authors
Please amend the paper according to the comments:
1. The title is incorrect as this study did not examine anxiety and depression. Please reconsider and add anxiety and depression symptoms.
2. Titles of the questionnaires should be written with capital letters, e.g., line 19: Generalized Anxiety Disorder Scale (GAD-7) etc.
3. Please clarify: The most frequently mentioned diagnoses were anxiety and depression, 6.1% and 5.4% respectively. Diagnoses were anxiety and depression, but based on what? These diagnoses can not be made by self-report screening questionnaires.
4. Typos, e.g., line 132: "with a Cronbach’s alpha than 0.80" etc.
5. 2.4. Ethical Aspects should be described in the procedure section.
6. The data in the abstract regarding the diagnosis of a mental health disorder are ambiguous. "When asked ‘Have you ever been diagnosed with any kind of mental disorder?’, the vast majority of students answered no; however, 9.5% reported a previous diagnosis of a mental disorder and, of these, 35.7% reported that it was diagnosed after the start of the COVID-19 pandemic." (Lines 200-202). These lines stated that this was based on the self-report question. So, the authors should pay a lot of attention to details in order not to provide overstatements.
7. Please calculate effect size for all analyses where possible.
8. Internal consistency reliability should be calculated.
9. A more pertinent analyses should be conducted like logistic regression, multiple regression analyses etc. The paper uses only univariate statistics which might be not enough for the current state of knowledge. Please enrich your paper by more nuanced approach with using multivariate analyses in predicting anxiety and depression symptoms.
Author Response
Responses to Comments from Reviewer 1
Response:
Thank you very much, we greatly appreciate your support. We believe that your thoughtful comments have greatly clarified, improved and strengthened the manuscript.
Please amend the paper according to the comments:
- The title is incorrect as this study did not examine anxiety and depression. Please reconsider and add anxiety and depression symptoms.
Response:
Thank you very much for your comments. Also, in view of the further analyses conducted, we changed the title to “Symptoms of anxiety and depression in Italian nursing stu-dents: prevalence and predictors”.
- Titles of the questionnaires should be written with capital letters, e.g., line 19: Generalized Anxiety Disorder Scale (GAD-7) etc.
Response:
Thank you very much for your comments. We made the corrections as you suggested.
- Please clarify: The most frequently mentioned diagnoses were anxiety and depression, 6.1% and 5.4% respectively. Diagnoses were anxiety and depression, but based on what? These diagnoses can not be made by self-report screening questionnaires.
Response:
Thank you very much for your comments. In this new version of the manuscript, we have removed the above sentence from the abstract and clarified it in the results section. The diagnoses of anxiety and depression were previously made by a specialist and do not relate to the data collected through the questionnaire. Specifically, the sentence now “The diagnoses most frequently self-reported by students, and which had previously been diagnosed by a specialist, were anxiety and depression, 6.1% and 5.4% respec-tively, with 6.8% of students reporting 2 or more diagnoses” (lines 219-221).
- Typos, e.g., line 132: "with a Cronbach’s alpha than 0.80" etc.
Response:
Thank you very much for your comments, we corrected them.
- 4. Ethical Aspects should be described in the procedure section.
Response:
Thank you for your suggestion. The authors have moved section 2.4. Ethical Aspects to section 2.2. Participants and Procedures, as you suggested.
- The data in the abstract regarding the diagnosis of a mental health disorder are ambiguous. "When asked ‘Have you ever been diagnosed with any kind of mental disorder?’, the vast majority of students answered no; however, 9.5% reported a previous diagnosis of a mental disorder and, of these, 35.7% reported that it was diagnosed after the start of the COVID-19 pandemic." (Lines 200-202). These lines stated that this was based on the self-report question. So, the authors should pay a lot of attention to details in order not to provide overstatements.
Response:
Thank you very much for your comments. In this new version of the manuscript, to avoid creating ambiguity we have removed the following sentence from the abstract: “The most frequently mentioned diagnoses were anxiety and depression, 6.1% and 5.4% respectively. Whereas, in the manuscript, we clarified that the diagnoses self-reported by the students had previously been diagnosed by a specialist (lines 219-221)
Thank you again for your suggestions
- Please calculate effect size for all analyses where possible.
Response:
Thank you very much. We believe that your thoughtful comments have greatly improved and strengthened the manuscript. In this new version of the manuscript, in Table 2, we have reported the effect size values calculated when the independent variable is dichotomous. As non-parametric statistical tests were performed, the effect size was calculated using the biserial rank correlation (1) and the better known Cohen's d (2). We hope we have met your expectations. Thank you again
- Internal consistency reliability should be calculated.
Response:
Thank you very much for your comments. The authors agree, we calculated the internal consistency in our sample, as you suggested. Our results show a Cronbach's Alpha value for the GAD-7 scale of 0.889 and 0.88 for the PHQ-9 scale (lines 138 and 147)
- A more pertinent analyses should be conducted like logistic regression, multiple regression analyses etc. The paper uses only univariate statistics which might be not enough for the current state of knowledge. Please enrich your paper by more nuanced approach with using multivariate analyses in predicting anxiety and depression symptoms.
Response:
Thank you very much, we appreciate your support. We believe that your thoughtful comments have greatly improved and strengthened the manuscript. In this new version of the manuscript, we conducted multiple binary logistic regression analyses to identify factors predictive of anxiety and depressive symptoms.
Reviewer 2 Report
Comments and Suggestions for Authors
Dear Authors,
the subject of the paper is actuality.
The abstract and the paper must formulate research questions or hypotheses.
What program did you use?
In the abstract, the result must be reformulate!
In the introduction write, and explain the students` clinical practice. It is an important topic and has a connection with anxiety!!!
Must formulate a title and the number of valid cases under figures!
Table 2 in 3.4. subchapter must explain better! Do you process ANOVA tests or correlations? How about the dependent and independent variables? Do you made a factor analysis? How about the statistical coefficients on the tables?
I wish you all the best!
Author Response
Responses to Comments from Reviewer 2
Dear Authors, the subject of the paper is actuality.
Response:
Thank you very much for your support. We are happy to know that you share the importance and topicality of the topic.
The abstract and the paper must formulate research questions or hypotheses.
Response:
Thank you very much for your comments. In this new version of the manuscript, we have added the research questions as you suggested. Specifically, we have added (lines 79-84):
" To this end, the following research questions were formulated:
1) Among Italian nursing students, what is the prevalence of anxiety and depres-sive symptoms?
2)What are the factors associated with anxious and depressive symptoms in Ital-ian nursing students?”
What program did you use?
Response:
Thank you for your suggestion. As was reported in section 2.5. Statistical Analysis, “Analyses were performed using IBM© SPSS Statistics v.25.0 and the statistical package Jamovi v.2.4.
In the abstract, the result must be reformulate!
Response:
Thank you for your suggestion. In this new version of the manuscript, we have reformulated the abstract as you suggested
In the introduction write, and explain the students` clinical practice. It is an important topic and has a connection with anxiety!!!
Response:
Thank you very much, we appreciate your support. In this new version of the manuscript, we have added more information regarding the link between anxiety and clinical training. Specifically, we have added (lines 53-55):
“Higher levels of anxiety during clinical training of nursing students have frequently been reported, mainly due to new or unfamiliar clinical settings, fear of committing mistakes, causing harm to patients and experiencing negative reactions [17,18].”
Must formulate a title and the number of valid cases under figures!
Response:
Thank you very much, the authors agree. We have reworded the titles and added information on the number of valid cases, as you suggested
Table 2 in 3.4. subchapter must explain better! Do you process ANOVA tests or correlations? How about the dependent and independent variables? Do you made a factor analysis? How about the statistical coefficients on the tables?
Response:
Thank you very much, we appreciate your support. In this new version of the manuscript, we defined the dependent and independent variables, specified the statistical coefficients and explained table 2 in more detail, as you suggested. Specifically, the text reads (lines 243-257):
“Table 2 shows the differences observed through Mann-Whitney-Wilcoxon or Krus-kal-Wallis tests. The dependent variables used in the analyses were the GAD-7 and PHQ-9 scores, while the independent variables included socio-demographic characteristics, such as age, son-in-law, etc.; students' perceptions of their own academic performance and socio-economic level; student-worker status; and drinking habits, such as cigarette smoking, cannabis use, alcoholic beverages, etc. As can be observed, significantly higher depressive symptom scores were encountered in students who were female (p=0.037), had a low perceived socio-economic level (p=0.024), had a previous diagnosis of mental disorders (p=0.006), had a low perceived academic performance (p=0.001), smoked tobacco (p=0.008), used cannabis (p=0.034) and consumed more than 4-5 alcoholic drinks on the same occasion (p=0.025). With regard to anxiety symptoms, they were significantly higher in students in a loving relationship (p=0.043), with a previous diagnosis of mental disorders (p=0.014), with a perceived low school performance (p=0.048), who smoked tobacco (p=0.032) and who consumed more than 4-5 alcoholic drinks on the same occasion (p=0.009).”
Reviewer 3 Report
Comments and Suggestions for Authors
This study recruited 148 nursing students to explore the prevalence of mental disorders among Italian nursing students after the majority of COVID-19 pandemic restrictions had been lifted. The findings revealed a high prevalence of anxiety and depression symptoms in this group. Factors such as romantic relationships, a previous diagnosis of mental disorders, perceived academic performance, tobacco use, cannabis use, and consuming more than 4-5 alcoholic beverages on a single occasion were associated with depressive and/or anxiety symptoms. While the study was well-conducted and the manuscript well-written, there were two major limitations: 1) the absence of a control group to determine whether nursing students have a higher prevalence of anxiety and depression symptoms compared to the general population, and 2) the cross-sectional design, which limited the ability to establish causal relationships between the COVID-19 pandemic and the observed variables.
Author Response
Responses to Comments from Reviewer 3
This study recruited 148 nursing students to explore the prevalence of mental disorders among Italian nursing students after the majority of COVID-19 pandemic restrictions had been lifted. The findings revealed a high prevalence of anxiety and depression symptoms in this group. Factors such as romantic relationships, a previous diagnosis of mental disorders, perceived academic performance, tobacco use, cannabis use, and consuming more than 4-5 alcoholic beverages on a single occasion were associated with depressive and/or anxiety symptoms. While the study was well-conducted and the manuscript well-written, there were two major limitations: 1) the absence of a control group to determine whether nursing students have a higher prevalence of anxiety and depression symptoms compared to the general population, and 2) the cross-sectional design, which limited the ability to establish causal relationships between the COVID-19 pandemic and the observed variables.
Response:
Thank you very much, we greatly appreciate your support. We believe that your thoughtful comments have greatly improved and strengthened the manuscript.
Based on the limitations you observed, in this new version of the manuscript we have included them as limitations of the study (lines 416-419 and 428-4429).
Thank you again
Round 2
Reviewer 1 Report
Comments and Suggestions for Authors
The paper has been improved significantly, great job! Thank you for comprehensive work on my comments.